# Molecular basis of specificity and deamidation of eIF4A by Burkholderia Lethal Factor 1

George W. Mobbs [1,7,9], Adli A. Aziz [1,2,9], Samuel R. Dix [1,9], G. M. Blackburn[1], Sveta E. Sedelnikova[1], Thomas C. Minshull[3,8], Mark J. Dickman [3], Patrick J. Baker [1], Sheila Nathan [4], Mohd Firdaus Raih[5] & David W. Rice [1,6 ✉]

*Burkholderia pseudomallei* lethal factor 1 (BLF1) exhibits site-specific glutamine deamidase activity against the eukaryotic RNA helicase, eIF4A, thereby blocking mammalian protein synthesis. The structure of a complex between BLF1 C94S and human eIF4A shows that the toxin binds in the cleft between the two RecA-like eIF4A domains forming interactions with residues from both and with the scissile amide of the target glutamine, Gln339, adjacent to the toxin active site. The RecA-like domains adopt a radically twisted orientation compared to other eIF4A structures and the nature and position of conserved residues suggests this may represent a conformation associated with RNA binding. Comparison of the catalytic site of BLF1 with other deamidases and cysteine proteases reveals that they fall into two classes, related by pseudosymmetry, that present either the *re* or *si* faces of the target amide/peptide to the nucleophilic sulfur, highlighting constraints in the convergent evolution of their Cys-His active sites.

[1] Krebs Institute, School of Biosciences, University of Sheffield, Sheffield S10 2TN, UK. [2] School of Biology, Faculty of Applied Sciences, Universiti Teknologi MARA Cawangan Negeri Sembilan, Kampus Kuala Pilah, 72000 Kuala Pilah, Negeri Sembilan, Malaysia. [3] Department of Chemical and Biological Engineering, University of Sheffield, Mappin Street, Sheffield S1 3JD, UK. [4] Department of Biological Sciences and Biotechnology, Faculty of Science and Technology, Universiti Kebangsaan Malaysia, 43600, UKM Bangi, Bangi, Selangor, Malaysia. [5] Department of Applied Physics, Faculty of Science and Technology and Institute of Systems Biology, Universiti Kebangsaan Malaysia, 43600, UKM Bangi, Bangi, Selangor, Malaysia. [6] Faculty of Science and Technology, Universiti Kebangsaan Malaysia, 43600, UKM Bangi, Bangi, Selangor, Malaysia. [7] Present address: Division of Chemistry and Chemical Engineering, California Institute of Technology, 1200 East California Boulevard, Pasadena, CA 91125, USA. [8] Present address: School of Molecular and Cellular Biology, Astbury Centre for Structural Molecular Biology, University of Leeds, Leeds LS2 9JT, UK. [9] These authors contributed equally: George W. Mobbs, Adli A. Aziz, Samuel R. Dix. ✉email: d.rice@sheffield.ac.uk

B. *pseudomallei* is an intracellular gram-negative bacterial pathogen of humans and animals[1] and the causative agent of Melioidosis[2,3]. This pathogen is found in moist soil and surface water[4] in Southeast Asia and Northern Australia where the disease is endemic[5]. More recently, cases of Melioidosis have been reported in China, India, Africa[6], and Bangladesh[7]. Melioidosis presents with a broad array of symptoms being commonly misdiagnosed as tuberculosis[8], typhoid fever, or malaria[9]. Typical symptoms include acute respiratory infection, acute bacteremia, abscess formation, or soft tissue infection with fever. Up to 85% of reported melioidosis cases are acute infections with the severity and outcomes of melioidosis very much dependent on risk factors such as bacterial load, route of infection and bacterial virulence factors. The fatality rate for melioidosis is between 10–50% and death from an acute infection can occur within 24–48 h after bacterial exposure, usually resulting from sepsis with or without pneumonia or localized abscesses[3]. A particular feature of B. *pseudomallei* is that, following infection, the organism is capable of lying dormant with a full-blown infection sometimes emerging decades later[10]. The ability of B. *pseudomallei* to persist in the environment[11] and to cause a devastating disease following a potentially long latency period after initial exposure[12], has raised concerns of its use as a bio-terror weapon[13] leading to its classification as a Tier 1 select agent[14].

Analysis of a number of microbial pathogens has shown that as part of their toxicity, they express enzymes capable of deami-dating the amide side chains of critical glutamine[15,16] or asparagine[17,18] residues disrupting key biological pathways in the host. Structural and biochemical studies have shown that, thus far, these toxins can be divided into two distinct families. The first family, the papain-like deamidases, have a fold similar to that of the cysteine protease, papain[16,19]. The second family is based on a fold similar to that of the C-terminal catalytic domain of the Cytotoxic Necrotizing Factor 1 (the CNF1-like deamidases)[15,20]. The latter fold is also shared by the multicellular eukaryotic N-terminal asparagine/glutamine deamidases which form part of the N-degron pathway[21] and also by the receptor-modifying deamidase, CheD[22]. B. *pseudomallei* Burkholderia Lethal Factor 1 (BLF1) is a member of the CNF1-like deamidase toxin family which acts by targeting the site-specific deamidation of Gln339 of the eukaryotic initiation factor 4 A (eIF4A)[23]. eIF4A is a member of the DEAD-box superfamily and is an essential component in the translation initiation complex, supplying RNA helicase activity responsible for melting the secondary structures present in mRNA prior to translation[24,25]. DEAD-box family members invariably contain two domains, an N- and a C-terminal RecA-like domain (residues 21–236 and 247–406, respectively in human eIF4A), connected through a flexible linker region[26]. The latter possesses an inherent degree of conformational plasticity as demonstrated by differences in the orientation of the RecA-like domains relative to each other in the closed (active) and open (inactive) states[27]. Gln339 is located in the C-terminal domain of eIF4A, between the conserved sequence motifs V and VI, a region believed to interface with RNA and ATP[28]. Site-specific deami-dation of eIF4A Gln339 results in extensive inhibition of protein synthesis in human cells[23] and further studies demonstrated that BLF1 is a key component for B. *pseudomallei* pathogenesis, with a Δ*BPSL1549* strain being 100-fold less virulent than the wildtype (WT)[23].

The structures of the complexes of several papain-like deami-dase toxins with their substrates have been solved to shed light on their specificity including the Cif-like homologs from B. *pseudomallei* (CHBP), *Yersinia pseudotuberculosis*, and *Photorhabdus luminescens* in complex with NEDD8[16,19]. However, notwith-standing the structure of the deamidase CheD and CheC, a mimic of its receptor substrate, and that of the complex between the human N-terminal asparagine deamidase (NTAN1) and a pep-tide substrate[21], currently there are no structures of a complex between any CNF1-like family toxin deamidase with its natural target and hence full details of the molecular specificity are unknown. In this paper, we present the structure of an inactive C94S mutant of BLF1 in complex with human eIF4A, allowing the mechanism and deamidation specificity of a CNF1-like glu-tamine deamidase toxin to be examined.

## Results

**Mechanistic similarities between BLF1 and the cysteine pro-teases.** A mechanistic similarity between catalysis by the cysteine proteases and the glutamine deamidase activity of BLF1 and CNF1 has previously been suggested given the similar orientation of a Cys-His dyad in their respective active sites[20,23]. In the cysteine protease papain, the Cys-His dyad is proposed to exist as a thiolate-imidazolium ion pair prior to the nucleophilic attack on the substrate by the thiolate anion to form the first anionic tet-rahedral intermediate[29–31]. This proposal is supported by bio-chemical and biophysical studies on free and Cys25 thiol-derivatives of papain using spectroscopic techniques, fluorometric titrations, kinetic deuterium isotope experiments, and proton nuclear magnetic resonance studies[32–36], as reviewed by Polgar & Halasz[37].

Despite there being only a 5% sequence identity between BLF1 and CNF1, their structures both adopt a common β-sandwich tertiary fold and can be superimposed with a 3.9 Å RMSD over 170 equivalenced α-carbon atoms[23]. Six of the conserved residues between these two toxins lie in their respective active sites, including a conserved L-S-G-C motif located on a short loop containing the putative cysteine nucleophile (BLF1: Cys94; CNF1: Cys866)[23]. In the structure of the C-terminal catalytic domain of CNF1 (C-CNF1 (PDB:1HQ0[20])), the active site cysteine occupies two positions, complicating analysis of its mechanistic role[20]. Moreover, investigations of the mechanism of the cysteine proteases, in general, are also hampered by oxidation of the active site cysteine, a feature that has been commonly noted (PDB:9PAP, 2CIO, 3P5U, 3P5V, 3P5X[38–40]). The presence of one or more oxygen atoms bound to the sulfur of the cysteine side chain clearly has the potential to alter the local geometry and therefore complicates an understanding of the interactions between the residues of the catalytic Cys-His dyad and the substrate. Whilst partial oxidation of Cys94 was a feature of the initial BLF1 WT structure (Fig. S1a) (PDB:3TU8[23]), more recently, we collected a new dataset to 0.99 Å, processing of which showed little evidence of radiation damage (Table 1) (PDB:6RVU). Analysis of the refined structure indicated that the sulfur had not been subject to the oxidation of its side chain, but the position of the sulfur atom of the cysteine side chain was otherwise unchanged (Fig. S1b). Furthermore, the multiple conformations of Ser92 that lie close to the active site in the partially oxidized structure were replaced by a single conforma-tion. In this reduced BLF1 WT structure, the $S_\gamma$ of Cys94 is 3.12 Å from the $N_{\epsilon2}$ of His106, consistent with a hydrogen bond between them. Analysis of the geometry of the hydrogen bond donor/acceptor atoms of the two residues indicates that the hydrogen bond has a more linear arrangement (D-H--A°) if the proton resides on the $N_{\epsilon2}$ of His106 (160°) rather than on the $S_\gamma$ of Cys94 (144°) (Fig. S1c, d). Thus, it is reasonable to deduce that His106 is protonated indicating the presence of a thiolate-imidazolium ion pair stabilized by hydrogen bonding and consistent with the "ion pair" proposal for the mechanism of papain, as discussed above. This situation contrasts with that seen in the serine proteases where a hydrogen bond is observed between the -OH of the serine

**Table 1 Data collection and refinement statistics.**

**Data collection**

| | BLF1 C94S: eIF4A $^{\Delta 20}$ (Form A) | BLF1 C94S: eIF4A $^{\Delta 20}$ (Form B) | BLF1 WT |
|---|---|---|---|
| Protein | BLF1 C94S: eIF4A $^{\Delta 20}$ (Form A) | BLF1 C94S: eIF4A $^{\Delta 20}$ (Form B) | BLF1 WT |
| Beamline | Diamond light source (UK) I04 | Diamond light source (UK) I03 | Diamond light source (UK) I03 |
| Wavelength (Å) | 0.9795 | 0.9763 | 0.9763 |
| Space group | C 2 | P 63 | P 21 21 21 |
| a, b, c (Å) | 135.9, 50.4, 95.7 | 199.5, 199.5, 51.3 | 37.3, 44.6, 116.9 |
| α, β, γ (°) | 90.0, 111.9, 90.0 | 90.0, 90.0, 120.0 | 90.0, 90.0, 90.0 |
| Resolution (Å) | 63.1–2.5 | 65.3 – 3.0 | 44.6 – 1.0 |
| $R_{pim}$ (I) [a] | 0.096 (0.441) | 0.015 (0.078) | 0.032 (0.242) |
| $<I>/<\sigma I>$ [a] | 8.1 (1.7) | 34.9 (9.0) | 12.6 (1.6) |
| Completeness (%) [a] | 99.2 (98.7) | 99.7 (97.9) | 86.7 (18.6) |
| No. of observations | 76647 | 198942 | 548111 |
| No. of unique reflections [a] | 20378 (1457) | 23808 (1702) | 93887 (1556) |
| Redundancy [a] | 3.8 (3.6) | 8.4 (8.4) | 5.8 (1.3) |
| CC$^{1/2}$ | 0.985 (0.703) | 0.999 (0.988) | 0.996 (0.793) |
| **Refinement** | | | |
| $R_{work}$ /$R_{free}$ | 0.20 / 0.28 | 0.19 / 0.23 | 0.13 / 0.15 |
| No. of non-H atoms | 4669 | 4716 | 2116 |
| Protein/Water | 4602 / 67 | 4716 / - | 1727 / 369 |
| Average B-factors (Å$^2$) (Protein/Water) | 27.6 / 19.4 | 95.0 / - | 11.3 / 25.7 |
| Bond length rmsd (Å) | 0.011 | 0.011 | 0.014 |
| Bond angle rmsd (°) | 1.68 | 1.55 | 1.80 |
| **Validation** | | | |
| Ramachandran favored (%) [b] | 95.4 | 94.4 | 97.6 |
| Additionally allowed (%) [b] | 4.6 | 5.1 | 2.4 |
| Outliers (%) [b] | 0.0 | 0.5 | 0.0 |
| Rotamer outliers (%) [b] | 2.0 | 4.5 | 0.0 |
| Cβ deviations (%) [b] | 0.4 | 0.2 | 0.0 |
| Clash score [b] | 6.22 (99th) | 7.77 (97th) | 1.16 (97th) |
| Molprobity score [b] | 1.89 (97th) | 2.30 (98th) | 0.94 (98th) |
| PDB ID | 7PPZ | 7PQ0 | 6RVU |

[a]Highest resolution shell is shown in parentheses
[b]Determined by Molprobity[67], with percentile scores ranked against structures deposited in the PDB within a ±0.25 Å range, see parentheses.

and the N$_{\epsilon 2}$ of the histidine in the catalytic triad where the histidine provides general base catalysis for the removal of the serine oxygen proton[41]. Subsequent discussions of the WT BLF1 toxin will refer to the coordinates for the fully reduced BLF1 structure (PDB:6RVU) which supersede those deposited earlier (PDB:3TU8).

**The structure of the BLF1 C94S:eIF4A$^{\Delta 20}$ complex.** We determined the structures of two crystal forms (A and B) of the catalytically inactive BLF1 C94S mutant in complex with human eIF4A$^{\Delta 20}$, at 2.5 and 3.0 Å, respectively (Table 1). These structures are closely related with BLF1 C94S binding between and forming interactions with, each of the N- and C-terminal RecA-like domains of eIF4A (Fig. 1a). Superposition of the coordinates of the complex for forms A and B, based on overlapping the structure of the toxin, reveals that the position of the C-terminal domain of eIF4A with respect to BLF1 is very similar (maximal Cα displacement 1.7 Å). The major difference between the two forms lies in the position of the N-terminal domain (maximal Cα displacement 3.8 Å) which can be seen to have rotated by an angle of ~4° against the BLF1 surface (Fig. S2a). Other differences between the two structures include the ordering of residues 135–150 in the N-terminal domain of eIF4A and residues 331–335 in its C-terminal domain, which are disordered in form A but which form a loop and an α-helix connecting β4 and β5 and a loop connecting β12 and β13 in form B, respectively. Further changes in the conformation of residues of eIF4A can also be observed in the loop between helices α1 and α2 (residues 50–55) and residues connecting β2 with α4 (77–86) in the N-terminal domain.

Comparison of forms A and B further reveals a shift in the position of the target Gln339 in eIF4A with respect to the Cys/Ser-His pair in the BLF1 active site. Whilst in form A the orientation of Gln339 is consistent with the suggested involvement of these residues in the mechanism and the expected nucleophilic attack on the side-chain amide of Gln339 by Cys94, in form B it is not. This arises as a result of the binding of an unmodelled buffer component in form B close to the carboxamide of Gln339 leading to a displacement in the position of the carboxamide group of Gln339 away from Cys94 by ~1 Å. We presume that this shift triggers the changes we observe between the two forms in the domain orientation of eIF4A (Fig. S2a). We note that a similar situation has previously been observed in a comparison of the structures of the complex between the papain-like Cif glutamine deamidase of *P. luminescens* (PDB:4FBJ[16]) and that of *Y. pseudotuberculosis* (PDB:4F8C[16]) with their substrate NEDD8, where the side chain of the target Gln40 in the former adopts a position that is inappropriate for catalysis[16]. We suggest that the orientation of Gln339 in form B represents a non-productive mode of binding and the following description will therefore focus on the interaction between BLF1 C94S and eIF4A seen in form A.

**Specificity of BLF1 involves interactions with both RecA-like domains of eIF4A.** A total surface area of 1900 Å$^2$ is buried in the formation of the complex by the interaction of two patches of residues on the surface of BLF1 with opposing faces of the N- and C-terminal domains of eIF4A across a wide groove between the two domains of the helicase (14 and 28 residues of BLF1 with 14 and 20 residues on the domains of eIF4A, burying 700 and

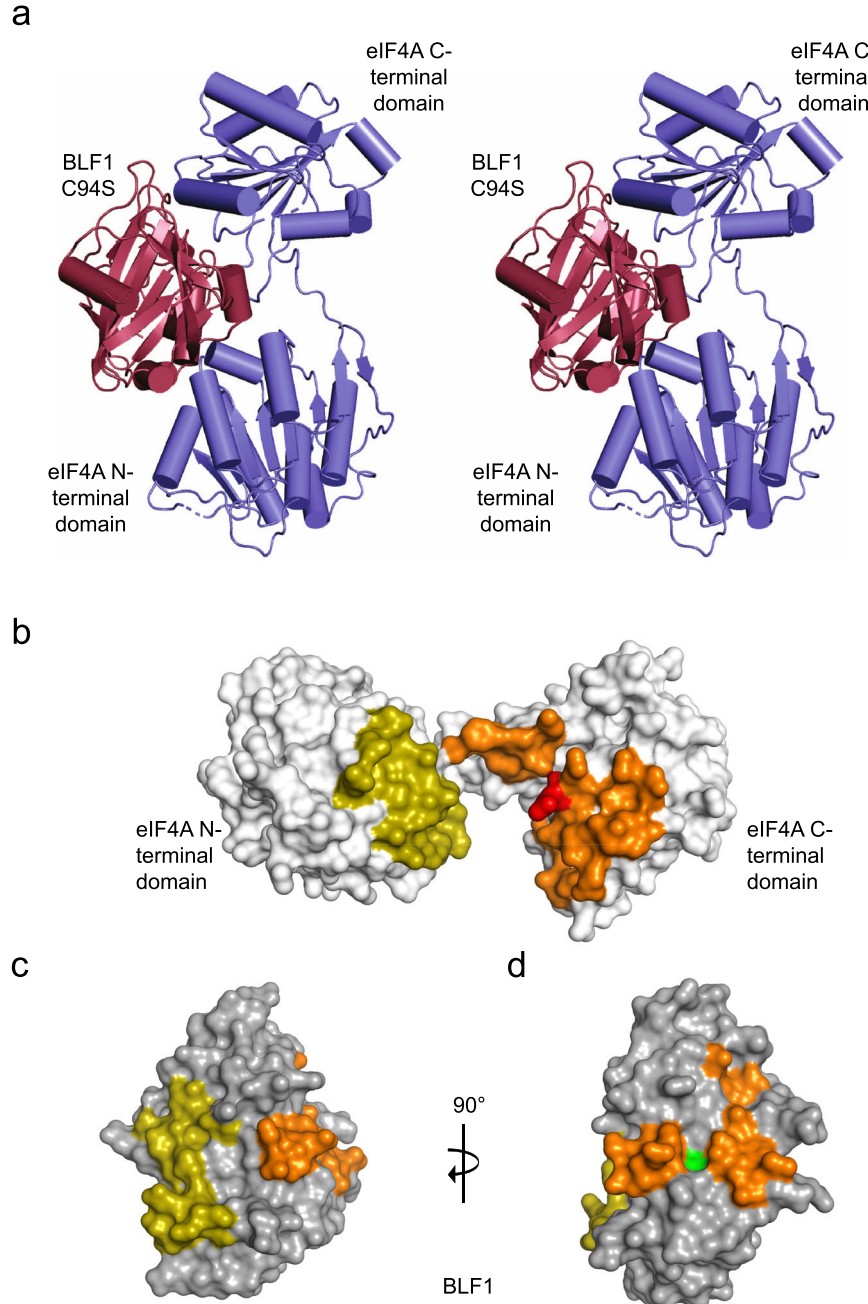

**Fig. 1 BLF1 C94S binds in the cleft between the two domains of eIF4A$^{\Delta 20}$. a** Stereo pair for cross-eyed viewing of the structure of the BLF1 C94S:eIF4A$^{\Delta 20}$ complex (PDB:7PQ0 (BLF1; maroon, eIF4A; blue)). **b** Surface representation of the N- and C-terminal domains of eIF4A. The two patches that interact with BLF1 C94S are shown in gold and orange, respectively with the remainder of the surface colored white. Gln339 of eIF4A is colored red. **c** Surface representation of the corresponding patches (gold and orange) on BLF1 that interact with eIF4A with the remainder of the surface colored gray. The view is rotated about the horizontal axis by 90° compared to the orientation of eIF4A shown in panel **b** so that the patches on BLF1 face the reader. **d** In a second view, modified by a rotation of 90° around the vertical axis, the surface of BLF1 flanking the active site is shown with the Cys/His pair colored green.

1200 Å$^2$ surface area, respectively) (Fig. 1b, c and Table S1a). Both of these patches involve the formation of a complementary network of largely polar interactions (Fig. S3a, b and Table S1b, c). We note the involvement of numerous negatively charged residues in BLF1 (Glu22, Glu70, Asp108, Glu124, Glu127, and Asp128), as a dominant feature of its interaction with positively charged side chains of eIF4A (Arg110, Arg161, Arg316, Arg362, and Arg365).

The interface between BLF1 and the C-terminal domain of eIF4A involves interactions with the loop containing the target

Gln339 of the latter together with residues flanking it, revealing that their interactions are mediated extensively by side chains (Fig. S3b). Those residues critical to the recognition include the side chains of Asp337, Gln339, and Gln340 in eIF4A which point outwards from its molecular surface, binding into the pocket provided by BLF1, where major contributions arise from the side chains of Trp66, Tyr89, Arg142, Gln145, and Asn168, and also the main chain of Cys/Ser94, Ser92, and Tyr90 (Fig. S3b and Table S1b, c). The amide nitrogen and oxygen atoms of the carboxamide of Gln339 in eIF4A are involved in

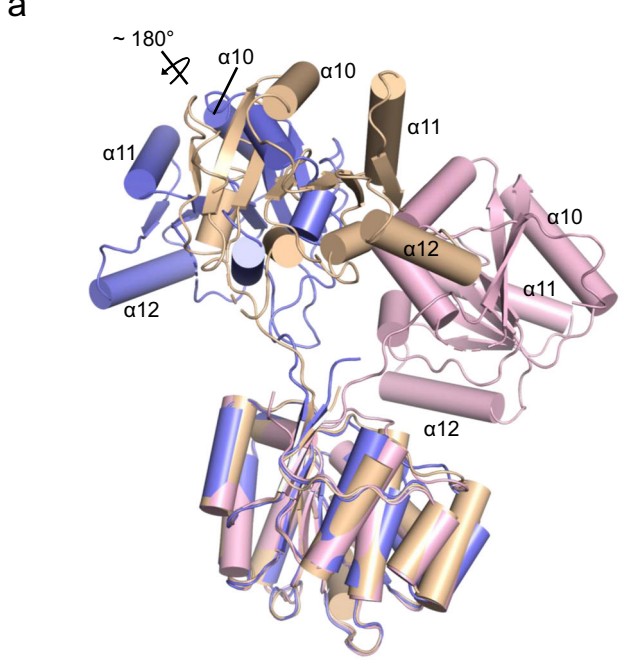

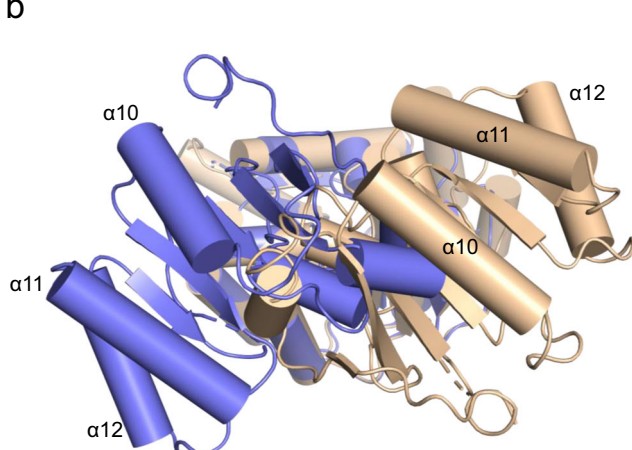

**Fig. 2 BLF1 C94S binds to a twisted conformation of the two eIF4A$^{\Delta20}$ domains.** Superposition of three different eIF4A structures on the basis of their N-terminal domains reveals the quite different positions of the C-terminal domain. Helices α10, α11, and α12 of eIF4A are labeled to enable the different orientations to be seen. **a** In the apo yeast eIF4A, (PDB:1FUU; wheat) the two domains adopt an open conformation whilst a complex of yeast eIF4A with eIF4G (PDB:2VSX; pink) reveals a closed conformation for eIF4A. Human eIF4A in complex with BLF1 (PDB:7PQ0; blue) adopts a different open conformation in which the eIF4A domains are twisted compared to apo. The approximate axis of rotation about which the C-terminal domain of eIF4A twists in the complex with BLF1 is indicated. **b** A view of the superposition between the N-terminal domains of apo yeast eIF4A and Human eIF4A in complex with BLF1 (background), shown in wheat and blue respectively, highlighting the approximate 180° rotation relating their C-terminal domains in the foreground.

hydrogen bonds with the carbonyl of Tyr90 and the π-electron system of Trp66 and with the main chain NH groups of Ser92 and Cys/Ser94, of BLF1, respectively (Fig. S3b). These interactions, together with the extensive interfaces formed by the two polar patches, serve to juxtapose the two molecules

placing the critical side chain of Gln339 in the active site of BLF1.

**BLF1 recognizes a twisted conformation of the N- and C-terminal domains of eIF4A.** Comparison of the structure of eIF4A in the complex with BLF1 C94S with other eIF4A structures in the PDB reveals a significant change to the relative orientation of its N- and C-terminal domains. The structures of the apo yeast eIF4A structure (PDB:1FUU[28]) and the active yeast eIF4A:eIF4G complex (PDB:2VSX[42]) differ with an open conformation of the two domains in the former and a closed conformation in the latter corresponding to a rotation of ~50°. However, their relative orientation in the structure of the complex between human eIF4A and BLF1 is dramatically altered with the two domains 10° more open compared to the yeast structure (PDB:1FUU) but, more importantly, twisted by 180° such that the two domains face each other quite differently (Fig. 2a, b). Whether this unusual conformation of eIF4A is solely a feature of its interaction with BLF1 or whether it reflects a conformation that the helicase adopts as part of its biological role is currently unclear. However, we note that more than 75 % of the residues in eIF4A buried in the interaction with BLF1 are strongly conserved across a representative set of eIF4A sequences (Table S1, Fig. S4, and Fig. 1b) suggesting that this conformation is relevant to the function of eIF4A. Moreover, the results of mutations to each of the three arginines in the HRIGRXXR motif (residues 358–365 inclusive) of eIF4A drastically reduce eIF4A cross-linking to RNA and abolish RNA helicase activity. This led to a model of eIF4A in which RNA binding was dependent on the HRIGRXXR region[43]. Our structure shows that two of these three arginines (Arg362 and Arg365) form part of one of the interacting surfaces that bind to one of the patches on BLF1. This surface on eIF4A then faces a positively charged region in the N-terminal eIF4A domain that interacts with the second patch on BLF1. Given that these two regions of eIF4A face each other, it seems plausible to suggest that the binding of BLF1 utilizes conserved surfaces on eIF4A that play a role in RNA binding.

Comparing the conformation of C94S BLF1 in the complex with the structure of WT BLF1 on its own reveals that small conformation changes can be seen in the contiguous patch on its surface which interacts with the C-terminal RecA-like domain of eIF4A. This includes interactions between residues in BLF1 from the adjacent loops between β7-β8 (residues Leu91-Cys/Ser94), the N-terminal end of β12 (Gln171), and the β13-β14 loop (residues Thr191-Ser197) with residues Arg319-Ser323 of eIF4A and the adjacent loop (Asp337-Gln340) containing the target Gln339 that is modified by the toxin (Fig. S2b). A consequence of these structural changes in BLF1 is an alteration to the torsion angle of Ser92 due to adverse steric interactions with Arg319 of eIF4A that slightly opens the specificity pocket in BLF1, thereby allowing the side chain of Gln339 to be accommodated. A further significant conformation change at the BLF1 active site between WT BLF1 and the C94S BLF1:eIF4A$^{\Delta20}$ complex is an alteration to the torsion angle of the side chain at residue 94. In WT BLF1 the cysteine at this position adopts a gauche$^+$ conformation, whereas the conformation of Ser94 in the mutant is gauche$^-$. The significance of this difference is discussed further below. Other small differences in the conformation of BLF1 in the complex compared to the wildtype appear to be due mainly to differences in crystal packing or related to differences in the solvent structure including the binding of small buffer ligands in the structure of one but not the other. We note that in the structure of the BLF1 C94S mutant on its own (PDB:3TUA[23]), two conformations of Ser92 (the conformation seen in WT BLF1 and that observed in the C94S mutant in complex with eIF4A) and

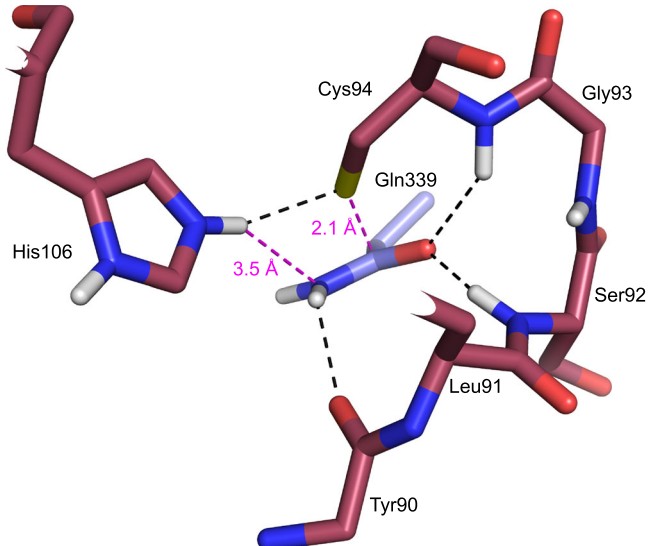

**Fig. 3 Active site of hybrid WT BLF1:eIF4A$^{\Delta20}$ model.** In the hybrid WT BLF1: eIF4A model, the eIF4a substrate Gln339 carboxamide forms hydrogen bonds to the main chain nitrogen of BLF1 Ser92 and Cys94 (black dashes) and lies close to the main chain carbonyl of Tyr90. His106 N$_{\varepsilon2}$ donates a hydrogen bond to the sulfur of Cys94 (black dashes) and lies 3.5 Å from the side chain amide nitrogen of Gln339 with the Cys94 sulfur 2.1 Å above the amide carbon (magenta dashes). The diagram was prepared using PyMOL (sulfur: yellow, oxygen: red, nitrogen: dark blue, carbon: (BLF1; maroon, eIF4A; blue).

conformational changes in the β13-β14 loop can also be seen, consistent with mobility in these regions being important in substrate recognition.

**Creating a hybrid model for the BLF1:eIF4A$^{\Delta20}$ complex.** Since the active sites of BLF1 (PDB: 6RVU) and the C94S mutant (PDB:3TUA) are very similar, it is reasonable to assume that the majority of the interactions seen between the enzyme and the substrate in the form A structure of the BLF1 C94S:eIF4A$^{\Delta20}$ complex resemble those in the fully active enzyme. However, as described above, the mutation does result in one significant change, namely an ~120° rotation from the gauche$^-$ torsion angle about the bond between C$_\alpha$ and C$_\beta$ of the side chain of the serine compared to the gauche$^+$ cysteine (Fig. S5a). This conformational change leads to the formation of a different pattern of hydrogen bonding in which the O$_\gamma$ of Ser94 acts as a hydrogen bond donor to the N$_{\varepsilon2}$ of His106 (2.8 Å) compared to that observed with the S$_\gamma$ of Cys94. In addition, a gauche$^-$ confirmation is not possible for the cysteine of the WT toxin because the sulfur atom would be too close to C$_\gamma$ of Leu91 (3.3 Å) resulting in adverse steric interactions, compared to the 3.7 Å distance observed for O$_\gamma$ of Ser94 (Fig. S5a). Analysis of the structures of the WT human N-terminal deamidase, NTAN1 (PDB:6A0E), and its inactive C75S mutant in complex with a peptide substrate (PDB:6A0H), reveal a pattern of torsion angles and torsion angle changes to the active site cysteine/serine which are identical to those seen in BLF1 and its C94S mutant in complex with eIF4A (Fig. S5b, c, d). We, therefore, presume that in the complex with the WT BLF1 toxin the cysteine adopts the most stable, anti-periplanar conformation of the side chain as seen in the structure of the fully reduced WT apoenzyme alone. Therefore, for the purpose of analysing the mechanism of BLF1 we have created a hybrid model of the complex between eIF4A and the wild-type toxin by superimposing the structure of the WT BLF1 onto the C94S

mutant in the complex and replacing the coordinates of Ser94 by Cys94, allowing this residue to adopt the conformation seen in WT BLF1. Hereafter, this model will be referred to as WT BLF1:eIF4A$^{\Delta20}$.

In the hybrid model of the WT BLF1:eIF4A$^{\Delta20}$ the sulfur of Cys94 lies 2.1 Å above C$_\delta$ of the substrate glutamine, Gln339, a distance that would be incompatible with a non-covalent interaction between them (Fig. 3). Equally, the relative orientations of the substrate amide and active site cysteine in an equivalent hybrid model in the NTAN1: peptide complex in which the torsion angle of Ser75 is replaced by that seen by the cysteine in the wild-type protein is clearly closely related to that observed in BLF1 but again with a steric clash between the cysteine sulfur and the asparagine amide (2.7 Å) (Fig. S5c). Therefore, it seems reasonable to assume that in the complexes of both WT BLF1 and NTAN1 this potential steric clash would be relieved by a small movement of the glutamine or asparagine side chain. For BLF1 this would leave the cysteine sulfur almost perpendicularly above the carboxamide plane of Gln339 being therefore well-placed to perform a nucleophilic attack on the amide carbonyl carbon atom. In addition, the carboxamide oxygen of the latter is located in the oxyanion hole, accepting two H-bonds from the main chain nitrogen atoms of Cys94 (3.2 Å) and Ser92 (2.6 Å) polarizing the double bond making the carbonyl carbon more susceptible to nucleophilic attack. Finally, the carboxamide nitrogen lies close to the main-chain carbonyl oxygen of Tyr90 (2.8 Å) and to the N$_{\varepsilon2}$ of His106 (3.5 Å) placing the latter in a good position to act as a proton donor to the departing nitrogen when the C-N bond is broken (Fig. 3).

**Comparison of the WT BLF1:eIF4A$^{\Delta20}$ complex with related cysteine deamidase/protease complexes.** Structural studies on BLF1 have previously highlighted the similarities in the putative catalytic machinery of the CNF1-like deamidases[23], the papain-like deamidases[16], the chymotrypsin-like cysteine proteases[44], and the papain-like cysteine proteases[45], all of which possess a Cys-His pair at their active sites. To compare the geometry of the catalytic machinery of the different classes of Cys/His deamidase/protease enzymes we sought to superpose their structures as observed in complexes with relevant substrates or substrate-like inhibitors. However, since these enzymes are based on three different folds, the four-layer α-β-β-α structure containing mixed beta-sheets of CNF1, the three-layer α-β-α structure of papain, and the two-domain antiparallel β-barrel fold of Tobacco Etch Virus (TEV) protease, there is no unequivocal way in which their structures should be superimposed. For example, they could be overlapped on the basis of their Cys-His pairs, or on the relationship of the nucleophilic cysteine to the substrate carbonyl attacked during catalysis, or on the polarity of the substrate carbonyl and the target C-N bond that is cleaved during the reaction catalyzed by their respective enzymes. We, therefore, explored superimposing the structures to preserve these different features.

**Comparison of the WT BLF1:eIF4A$^{\Delta20}$ complex with chymotrypsin-like cysteine proteases.** Superposition of the active sites of the WT BLF1:eIF4A$^{\Delta20}$ complex with the wild-type chymotrypsin-like cysteine protease from TEV in complex with a product of the reaction (PDB:1LVM[44]) revealed that, despite the differences in the fold, it was possible to superimpose the respective cysteine side chains, both of which are in the gauche$^+$ conformation (Fig. 4a, b). Similarly, the active site cysteine of the chymotrypsin-like protease from SARS Cov-2, 3CL$^{Pro}$, in complex with an α-ketoamide inhibitor (PDB:6Y2E, 6Y2F[46]) is also in the gauche$^+$ conformation[46]. The superposition of WT

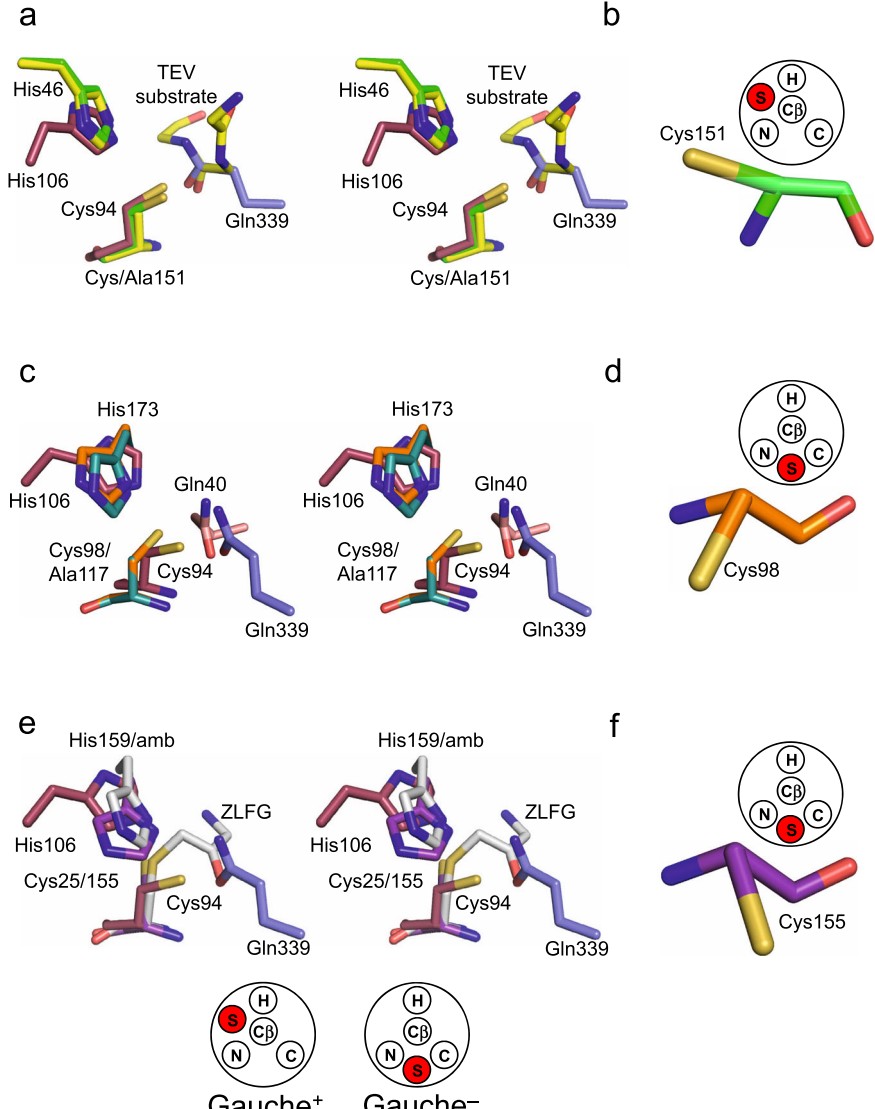

**Fig. 4 The relative positions of the substrate and catalytic cysteines in WT BLF1:eIF4A$^{\Delta 20}$ model compared to other structures. a** Stereo view of the active site of WT BLF1:eIF4A$^{\Delta 20}$ complex (BLF1; maroon, eIF4A; blue) superimposed with that of the TEV C151A protease with its substrate (PDB:1LVB; yellow), and with the WT TEV protease (PDB:1LVM; green). The target carboxamide group of the side chain of Gln339 of eIF4A (blue) superposes well with the scissile peptide of the TEV substrate (yellow). The superposition shows that the nucleophilic cysteine of TEV and BLF1 approach the *re* face of their respective substrates. **b** Cys151 of TEV showing it adopts the gauche$^+$ conformation. **c** Stereo view of the local superposition between the WT BLF1:eIF4A$^{\Delta 20}$ complex (BLF1; maroon, eIF4A; blue) and the Cif C117A:NEDD8 complex (PDB:4F8C (Cif; teal, Nedd8 substrate; pink)), and AvrPphB (PDB:1UKF; orange), maximizing the overlap of their Cys/His pairs. The superposition shows that the nucleophilic cysteine of Cif approaches the *si* face of Gln40 Nedd8 substrate, compared to the *re* face seen in the BLF1:eIF4A$^{\Delta 20}$ complex. **d** Cys98 of the Cif-like protease AvrPphB showing it adopts a gauche$^-$ conformation. **e** Stereo view of the WT BLF1:eIF4A$^{\Delta 20}$ complex (BLF1; maroon, eIF4A; blue) superimposed with the papain-ZFLG inhibitor complex (PDB:1KHP; white), and with the papain-like protease, Amb an 11 (PDB:5EF4; purple) revealing the common position of the oxyanion. **f** Cys155 of the papain-like protease, Amb an 11 showing it adopts a gauche$^-$ conformation.

BLF1:eIF4A$^{\Delta 20}$ with the complex of the inactive C151A TEV mutant with its peptide substrate (PDB:1LVB[44]) also maintains an overlap of the imidazole rings of the active site histidines and the δ-carboxamide of Gln339 in eIF4A with the TEV substrate peptide (Fig. 4a). An equivalent overlap can be achieved between the complex of NTAN1 and its peptide substrate and TEV given the similarity of the former to the BLF1:eIF4A$^{\Delta 20}$ complex. Therefore, not only is the relationship of the nucleophilic cysteine to the substrate carbonyl preserved but also the polarity of the substrate carbonyl and target C-N bond across the active site is maintained in both cases with the *re* face of the substrate amide/peptide towards the attacking cysteine (Fig. 4a). The main consequence of the difference in the fold between the two enzymes is

that while the imidazole ring of the active site histidine in WT TEV (His46) occupies the same space as that of His106 in BLF1, the main chain atoms of these two histidines do not overlap and the histidine approaches the cysteine sulfur from a different direction (Fig. 5). Nevertheless, despite this difference in structure, the hydrogen bond between the histidine and the cysteine uses the same nitrogen (N$_{\varepsilon 2}$) on the imidazole ring (Fig. 4a). Therefore, these two enzymes share remarkably close similarities of their catalytic machinery and the positions of their target amide/peptide bonds.

**Comparison of the WT BLF1:eIF4A$^{\Delta 20}$ complex with papain-like cysteine deamidases/proteases**. Manual superposition of the

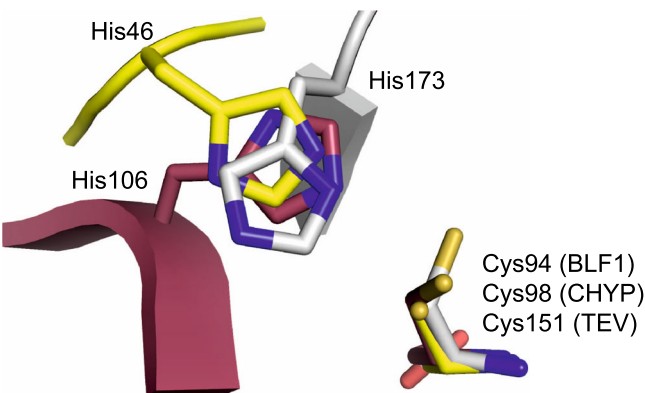

**Fig. 5 Superposition of the catalytic histidine from the three families.**
Local superposition of the catalytic histidine and cysteine residues from
BLF1 (Cys94/His106; maroon), TEV (Cys151/His46; yellow), and CHYP
Cif/ AvrPphB (Cys98/ His173; white). In both BLF1 and TEV, the imidazole
rings occupy the same approximate position and form a hydrogen bond
with their respective cysteine residues using the same, $N_{\varepsilon 2}$, Nitrogen,
despite differences in the position of the main chain. In contrast, in Cif, the
main chain approaches from a third location, to leave the imidazole ring of
His173 occupying a similar position but utilizing the nitrogen $N_{\delta 1}$ to
generate a hydrogen bond with its catalytic cysteine.

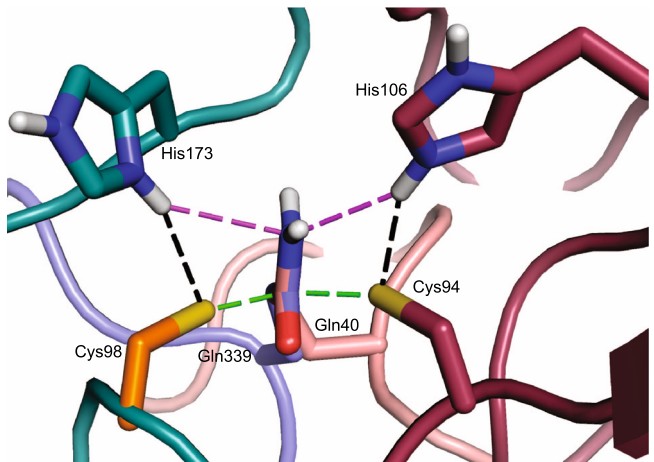

**Fig. 6 Comparisons of the active site architectures between deamidases
attacking the re and si face of the target amide.** A superposition between
complexes of the WT BLF1:eIF4A$^{\Delta 20}$ hybrid model (BLF1; maroon, eIF4A;
blue) and Cif C117A:NEDD8 complex (PDB:4F8C (Cif; teal,
NEDD8 substrate; pink))/ Cys98 of AvrPphB (PDB:1UKF; orange). The
superposition is based on the glutamine side chain substrates to show the
catalytically active Cys-His pairs are positioned on opposite faces of the
amide substrate in a pseudo-mirror plane arrangement. Hydrogen bonds
between the histidine imidazolium ring and the cysteine thiolate are shown
in black dashed lines, with the distances between the thiolate and glutamine
carboxamide carbon highlighted in green, and the imidazolate $N_{\varepsilon 2}$ hydrogen
and the glutamine carboxamide nitrogen in magenta.

active sites of the papain-like glutamine deamidase CHYP C117A
mutant (Cif-like homolog in *Y. pseudotuberculosis*) in complex
with NEDD8 (PDB:4F8C) with WT BLF1:eIF4A$^{\Delta 20}$ enables
superimposition of the side chains of their respective active site
histidines and the Cα-Cβ bond of their cysteine/alanine residues
(Fig. 4c). Similarly, as observed above with the superposition of
BLF1 and TEV, again in the superposition of BLF1 and CHYP,
because of the difference in the fold, while the two imidazole rings
of the two histidines overlap, their main chain atoms do not

(Fig. 5). In CHYP the position of the sulfur of the active site
cysteine has not been observed directly since a structure of the
wild-type enzyme is not available. However, it can be inferred
from the structure of the related papain-like protease, AvrPphB,
(PDB:1UKF, 3.6 A$^2$ r.m.s.d[47]) in which the cysteine has the
alternate gauche$^-$ torsion angle and the hydrogen bond between
the cysteine and histidine involves $N_{\delta 1}$ rather than $N_{\varepsilon 2}$, suggesting
that an equivalent interaction is made in CHYP (Fig. 4d). In
comparing the WT BLF1:eIF4A$^{\Delta 20}$ and CHYP C117A:NEDD8
complexes, while the N and the O atoms of the amide group of
the substrate glutamine Gln40 in NEDD8 superimpose on those
of Gln339 of eIF4A, the scissile C-N bond of the latter does not
overlap with the amide bond of Gln40 of NEDD8. This situation
arises as a result of the fact that in CHYP the cysteine approaches
the substrate carbonyl on its *si* face rather than the *re* face as in
eIF4A and essentially the polarity of the substrate as it crosses the
active site is reversed (Fig. 4c). Whilst in this situation a good
overlap of the functional elements of the respective catalytic
cysteines and histidines of these two enzymes, together with the
maintenance of the position of the substrate oxygen in the oxy-
anion hole is observed (Fig. 4c), a second completely different
superposition can be envisaged. In this second superposition, the
polarity of the substrates is maintained but only at the cost of no
longer overlapping the Cys-His pairs which now lie on opposite
faces of the amide substrate in CHYP compared to BLF1 in a
pseudo-mirror plane arrangement (Fig. 6). The adoption of the
gauche$^+$ (BLF1) or gauche$^-$ (papain) conformation of the cysteine
side chains is a conformational feature that maintains the pseu-
dosymmetric arrangement of substrate, histidine, and cysteine in
their respective active sites. In both cases, this facilitates the
alignment of a sulfur lone pair with the π* (vacant antibonding)
orbital of the carbonyl carbon while maintaining the ability of the
cysteine sulfur to ion pair with the imidazolium ring. While in
these structures the histidine N-H bond is primarily orientated
towards the sulfide, the arrangement is such that an oscillation of
the plane of the imidazole ring would direct a proton towards the
amide amino group thereby providing general acid catalysis for
its departure upon breakage of the C-N bond.

The overlap of the WT BLF1:eIF4A$^{\Delta 20}$ complex with
complexes of the papain-like proteases follows a pattern that is
very similar to the overlap with the papain-like deamidases as, in
the papain-like proteases, the nucleophilic cysteine approaches
the *si* face of the peptide. This can be seen, for example, in the
covalent complex of papain with a diazomethylketone inhibitor
(PDB:1KHP[45]) from the directionality of the substrate-like
portion of the inhibitor as it approaches the active site together
with the relative positions of the cysteine and the peptide-like
mimic of the inhibitor (Fig. 4e). Equally, as seen in the papain-
like deamidases, papain itself, and the Amb an 11 papain-like
protease (PDB:5EF4[48]) the cysteine side chain is also found in the
gauche$^-$ conformation forming a hydrogen bond with the active
site histidine $N_{\delta 1}$ atom of the imidazole ring (Fig. 4f).

Historic mechanistic proposals of the cysteine proteases
envisaged a stable tetrahedral intermediate in a classical, two-
step addition-elimination process[30,31]. The detailed similarities in
the arrangements of the catalytic groups of the different classes of
deamidase/protease enzymes highlighted in this work support the
suggestion that their mechanisms are similar. Therefore, based on
the proposed mechanism for papain, the role of the Cys94 in
BLF1 is to act as a nucleophile with its anionic sulfur attacking
the amide carbonyl together with general acid catalysis from
His106 to the nitrogen leaving group as part of the enzyme
catalytic cycle. Some computational studies have suggested that
the mechanism of papain proceeds via a single-step displacement
reaction at a trigonal center where the departure of nitrogen
is concerted with bonding to sulfur, originating with Peter

Kollman's work in 1990[49] and leading to a definitive analysis of the cruzain cysteine protease by Ferrer and Moliner[50]. As with such cysteine proteases, our structural analysis of the BLF1:eIF4A$^{\Delta 20}$ complex cannot resolve whether the mechanism of deamidation follows a single-step displacement or a stable tetrahedral intermediate in a two-step addition-elimination process.

## Discussion

The discovery that *B. pseudomallei* BLF1 is a toxic agent capable of blocking initiation-dependent protein synthesis in eukaryotes by deamidation of critical glutamine in the eIF4A RNA helicase provided insights into the pathogenicity of this organism[23]. The results described here highlight the molecular factors that control the specificity of the BLF1 toxin for its eukaryotic target, revealing the complementary nature of two patches on the surface of BLF1 that interact with conserved regions on the opposing RecA-like domains of eIF4A. However, the structure of the BLF1:eIF4A$^{\Delta 20}$ complex shows that the two RecA-like domains are held in a completely different orientation to each other to those seen previously. Taken together with results elsewhere that implicate two arginine residues involved in the interactions with BLF1 as playing a role in the recognition of RNA by eIF4A, this suggests that the toxin may be exploiting some elements of the structure of eIF4A that provide the binding site of its RNA substrate. In turn, this opens the possibility that there are further conformational changes involved in the action of eIF4A that are yet to be defined.

Given the high mortality rates associated with *B. pseudomallei* infections, the development of new therapies is critical[3,51,52]. In other work, an understanding of the mechanism of cysteine proteases has been important in the development of inhibitors for this family of enzymes[53] including, more recently, nirmatrelvir, the Pfizer inhibitor of the coronavirus cysteine protease, 3CL$^{pro}$, thus re-emphasizing the potential value of such mechanistic studies (PDB:7SI9)[54]. Despite their difference in fold, the close similarity of BLF1's catalytic machinery to the papain-like and chymotrypsin-like cysteine proteases and to the papain-like deamidases demonstrates their global equivalence and represents an interesting example of the way in which these enzymes have evolved. The similarities in structure and active site layout of the papain-like proteases and deamidases are clearly suggestive of the operation of a divergent evolutionary pathway from a common ancestor to generate these related chemistries. In contrast, while mechanistic similarities are evident in the comparisons between the BLF1-like deamidases and the chymotrypsin-like cysteine proteases, their difference in fold argues for this being the result of a process of convergent evolution to common catalytic machinery. All of these enzymes are designed to perform the chemically similar process of a nucleophilic attack to cleave a C-N bond on either the *re* face or the *si* face of their respective substrates. The sub-structural similarities we have identified provide a unifying feature across the different fold families thus allowing their mechanistic relationships to be understood. Moreover, it is interesting to note that despite the unrelated folds of these different classes of protease/deamidase enzymes, there appear to be consistent differences in the observed conformation of the active site cysteine, which correlates with the direction of attack on the carbonyl by the nucleophilic sulfur. Whether this arises from an aspect of the underlying chemistry or some restriction in the underlying folds is, as yet, unclear and may require the discovery of further examples of proteases/deamidases for the resolution of this issue. While the low sequence identity between disparate members of the family has been an obstacle in identifying new family members, the recognition of remote homologs through a combination of the analysis of sequence motifs and developments in protein fold prediction[55] will undoubtedly uncover more enzymes of this type and facilitate a deeper understanding of the molecular basis of their catalysis and substrate recognition. In turn, such information is likely to be helpful in the design of new inhibitors including compounds that might be useful in the treatment of melioidosis.

## Methods

**Protein expression and purification**. An inactive mutant of BLF1, C94S, was expressed in *E. coli* BL21 (DE3) cells (Novagen) grown in LB broth supplemented with 100 mg l$^{-1}$ Ampicillin, with expression induced by the addition of 1 mM IPTG at an OD$_{600}$ of 0.6 before incubation at 37 °C for 4 h. A construct of human eIF4A missing 20 residues at the N-terminal end, eIF4A$^{\Delta 20}$, was expressed in *E. coli* Tuner (DE3) cells (Novagen) in LB broth supplemented with 50 mg l$^{-1}$ Kanamycin, grown to OD$_{600}$ of 1 then chilled to 4 °C for 30 min on ice before induction with 1 mM IPTG and incubation at 13 °C for 72 h. Prior to purification, the *E. coli* cells (2 g dry weight) were stored on ice in 20 ml of a 50 mM TRIS pH 8 buffer, lysed using three rounds of sonication at 16 microns for 20 s using a Soniprep 150, and the lysate clarified by centrifugation at $60,000 \times g$ for 20 min. The clarified lysate was loaded onto a 5 ml DEAE FF column (GE Healthcare) and eluted with a 50 mM TRIS buffer at pH 8 using a linear gradient from 0–0.3 M NaCl across 8 column volumes. Fractions containing BLF1 were combined before concentration using a vivaspin (10 kDa) to a volume of 2 ml and further purified on a Superdex 200 16/60 size exclusion column (GE Healthcare). His-eIF4A was purified by affinity chromatography using a nickel column Ni-HP 5 ml (GE Healthcare) eluted with a 50 mM TRIS buffer at pH 8 with 0.5 M NaCl and a linear gradient of 0–0.5 M imidazole over 10 column volumes. To form a complex, a 4:1 molar ratio of BLF1 C94S to eIF4A$^{\Delta 20}$ was mixed in a 50 mM TRIS buffer at pH 7.5 with 0.1 M NaCl. The BLF1 C94S:eIF4A$^{\Delta 20}$ complex was separated from excess BLF1 by gel filtration using a Superdex 200 16/60 size exclusion column. The fractions containing eIF4A and BLF1 run with an apparent molecular weight of ~70 kDa consistent with the formation of a stoichiometric 1:1 complex. Subsequent in-gel tryptic digestion of eIF4A in conjunction with mass spectrometry analysis confirmed that the C94S mutant was inactive with no deamidation of Gln339 observed in contrast to treatment with the WT toxin (Figure S6). The BLF1 C94S:eIF4A$^{\Delta 20}$ complex was then buffer exchanged into a 10 mM TRIS pH 7.5 buffer and concentrated to 12 mg ml$^{-1}$ for crystallization trials.

**Crystallization structure determination and analysis**. Two distinct crystal forms of BLF1 C94S:eIF4A$^{\Delta 20}$ were grown at 17 °C in sitting drop trials containing 200 nl of a 0.1 M MES pH 6.5, 50 mM MgCl$_2$, 10% (w/v) 2-propanol, 5% (w/v) PEG 4000 solution and 500 nl purified complex at 12 mg ml$^{-1}$ (form A); or 200 nl of a 0.1 M HEPES pH 7.5, 4% (w/v) PEG 6000 solution mixed with 200 nl purified complex at 12 mg ml$^{-1}$ (form B). Crystals were flash-cooled with liquid nitrogen in mother liquor to which 30% (v/v) ethylene-glycol had been added as a cryoprotectant. X-ray diffraction data were collected at the Diamond Light Source (UK) on beamline I04 (form A) or I03 (form B) to a resolution of 2.5 and 2.24 Å, respectively. The data were handled using the CCP4 suite[56] and processed by Xia2[57], XDS[58], and AIMLESS[59]. Although the diffraction of crystal form B extended to 2.24 Å, the observation of anisotropy led to a decision to limit the resolution to 3 Å. The two crystal forms were solved independently but using a similar approach. For crystals of form A, initial phases were calculated following molecular replacement in PHASER[60] using coordinates for the C94S mutant of BLF1 (3TUA), the N-terminal domain of eIF4A (PDB:2ZU6 residues 63-231[61] and the C-terminal domain of eIF4A (PDB:2ZU6 residues 240–389) as three separate search ensembles. For crystals of form B, the coordinates for the N-terminal domain of eIF4A from PDB entry 2G9N (20-238) replaced those from 2ZU6 in an equivalent molecular replacement search. Model building was carried out in an iterative fashion using Coot[62] and the structures were refined using PHENIX[63] (form A) or REFMAC[64] (form B). Crystals of WT BLF1, isomorphous to those grown as previously described[23], were cryoprotected and data collected to 0.99 Å on beamline I03 at the Diamond Light Source. The structure was refined using REFMAC. Details of the structure solution and refinement are provided in Table 1. The nature of the interface formed between BLF1 C94S and eIF4A in the complex was analysed using PDBePISA[65]. The buried surface areas reported are rounded to the nearest 100 Å$^2$. Sequence alignment figures were generated using ESPript3[66].

**Mass spectrometry analysis**. LC ESI MS analysis was performed as previously described[23]. In brief, following in-gel tryptic digestion peptides were analysed using LC ESI MS analysis using a maXis UHR TOF mass spectrometer (Bruker Daltonics) using an automated acquisition approach. MS and MS/MS scans (m/z 50–3000) were acquired in positive ion mode. Lock mass calibration was performed using HP 1221.990364. Line spectral data were then processed into peak list by Data analysis (Bruker Daltonics) and the sum peak finder algorithm was used for peak detection using a signal to noise (S/N) ratio of 10, a relative to the base peak intensity of 0.1%, and an absolute intensity threshold of 100. Extracted ion chromatograms were generated for both the non-deamidated and deamidated peptide in Data analysis. MS/MS spectra of the peptides identified were manually verified.

**Reporting summary**. Further information on research design is available in the Nature Research Reporting Summary linked to this article.

## Data availability

Images used in data processing are available upon request. The atomic coordinates and structure factors generated in this study are available at the PDB with accession codes, 6RVU, 7PPZ, and 7PQ0.

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

## Acknowledgements

A.A.A. acknowledges a Ph.D. scholarship from Majlis Amanah Rakyat (MARA) Malaysia, M.J.D. acknowledges support from the Biotechnology and Biological Sciences Research Council UK (BB/M012166/1), S.R.D., S.N., M.F-R., P.J.B., and D.W.R. acknowledges support from a Royal Society International Collaboration Award (IC170306). We would also like to thank Diamond Light Source (DLS) for the time on beamlines I03 and I04, station staff for assistance with data collection, and STFC for DLS access and funding (MX8987 and MX17773).

## Author contributions

G.W.M., D.W.R., S.N., and P.J.B., designed the research. G.W.M., S.E.S., A.A.A., and S.R.D., performed the experiments and analysed the data. T.C.M. and M.J.D. performed the mass spectrometry. G.W.M., A.A.A., S.R.D., M.F-R., D.W.R., S.N., G.M.B., and P.J.B. wrote the paper.

## Competing interests

The authors declare no competing interests.
