## [Peer Review File · Communications Biology]

Reviewers' comments:

Reviewer #1 (Remarks to the Author):

The manuscript

Molecular basis of specificity and deamidation of eIF4A by Burkholderia Lethal Factor 1

by Mobbs, Aziz, Dix and colleagues describes protein crystallographic work on a complex of Burkholderia Lethal Factor 1 (BLF1) with eukaryotic initiation factor 4A (eIF4A). BLF1 deamidates Q339 of eIF4A, which leads to inhibition of protein synthesis, a major aspect of the pathogenesis of *B. pseudomallei* in Melioidosis. BLF1 is a cysteine-dependent deamidase and shows structural similarity to the deamidase domains of cytotoxic necrotizing factors (CNFs) despite extremely low sequence homology. A crystal structure of BLF1 has been published by the same authors about 10 years ago, but this previous study did not provide insight into substrate/eIF4A binding. The previous structure displayed oxidation of the active site cysteine nucleophile, which would probably have hampered complex formation.

The authors now close this gap by providing an ultra-high resolution crystal structure of the reduced apo-form of BLF1 together with two medium resolution structures of an inactive variant of BLF1 (C94S) in complex with eIF4A. Together with other crystal structures of eIF4A from yeast (the authors use human eIF4A in this new study), the complex structures shown here demonstrated a structural flexibility with respect to the relative arrangement of the two RecA-like domains of eIF4A. Further, because the CNF-like deamidases show mechanistic similarity to the structurally unrelated papain-like proteases (by using a similar cysteine-histidine couple to perform nucleophilic attack on their substrates), the authors extend their structural analysis by inspecting the attacking angles of the nucleophile and drawing interesting conclusions about the relative arrangements of residues participating in the catalytic process.

This is an interesting and insightful manuscript that dwells exclusively on structural data. If I have one general criticism about this work it would be that the structural analysis has not led to further experimentation to corroborate conclusions drawn from the structure, not necessarily by performing laboratory experiments but potentially by using high-level computations such as MM/QM simulations to translate the more static crystallographic pictures obtained in this study into trajectories that could provide even more insight into the mechanism of CNF-like deamidases. However, given that this manuscript is a communication and knowing how costly and time-consuming (and frustrating) such calculations can be, I am convinced that this manuscript is a valuable contribution as it stands.

Some points that the authors may want to check are:

Line 32: Is the comma after proteases correct?

L118-121: The authors speculate about the protonation state of H106/C94, which they base on geometric considerations. Of course, the very high resolution structure allows for such speculations. There are two questions here: could the authors perform calculations to determine the protonation state of the respective residues (I remember that these are not too difficult to do), and can they (maybe just in a rebuttal letter) provide a bit more insight into how they have refined the high resolution crystal structure of apo BLF1, since refining against such data can provide a challenge in itself but may actually lead to direct observation of protons if one is lucky? L216-217 sound slightly strange, maybe there should be a comma after slightly (this referee is not a native speaker)

L238: the comma after because seems wrong

L281: using just "TEV" to refer to "TEV protease" seems to be lab jargon – consider revising to be clear about what is meant

L376-377: the authors speculate that BLF1 mimics binding of RNA to eIF4A – are there structures to support this?

In table 1, I am missing information about CC1/2

Stereo figures: I suggest changing from wall-eyed to cross-eyed layout. This may be a subjective suggestion, but I have difficulties to perceive figures in this manuscript in stereo (I double-checked with both options in PyMOL to write this report, cross-eyed seems better (for me at least)). I also

have the impression that the labels within the stereo figures are not adjusted to the depth of the figure, this could be improved.

L580: refinement of the apo structure – I wonder if REFMAC will really give the best results here, have the authors also tried other programs (SHELXL etc)?

Reviewer #2 (Remarks to the Author):

The Authors present a manuscript describing a crystal structure of a complex between an inactive C94S mutant of BLF1 and human eIF4A. The authors intensively analyzed mechanistic similarities of BLF1 to cysteine proteases and carefully elucidated two crystal structures of BLF1 in different forms/space groups which interestingly suggest the orientation of the substrate Gln in different (productive/non-productive) modes. The authors then investigated interactions of BLF1 with eIF4A using a BLF1 structure in form A that is catalytically inactive but mimics a productive mode of the enzyme. Of note, the conformational change in the beta13-beta14 loop is of great interest, which supports the mobility of these regions in substrate recognition. Then, the authors generated a hybrid model for the BLF1:eIF4A Δ 20 complex and thoroughly analyzed the hybrid model with related cysteine deamidase/protease complexes. In conclusion, this manuscript highlights the molecular factors for the specificity and deamidation of eIF4a by BLF1. Overall, I found the manuscript interesting and well written with informative/essential figures. The structural data was described in detail and the claims are appropriate. However, I have some minor comments that should be addressed prior to publication.

1. Please provide broader meaning/speculation of this research at the end of the abstract.
2. As a non-native but scientific-English-fluent scientist, this reviewer asks the authors to revise complex sentences and relatively long sections to make them more clear and intuitive in meaning. Some of them are as follows; line 62-67 (too long and complex for readers to easily follow and understand), line 87-89, line 161-227 (could be divided into two sections), line 233-237 (too long and complex for readers to easily follow and understand), line 271-368 (could be divided into 2~3 sections), line 307-310.
3. In the beginning of Introduction, brief/short description of Melioidosis symptoms and pathogenic severeness would be informative for readers to be able to roughly understand the disease without further consideration.
4. Line 38: man  human
5. Line 66: one another  each other(???)
6. Line 97: between BLF1 and CNF1, their structures commonly adopts a beta-sandwich fold and could be well superimposed with r.m.s.d. of ?? angstrom among all the Calpha atoms (This part could be revised something like this).
7. Line 135-137: Figure S2A could be placed in the manuscript since it shows the major difference between two forms.
8. Line 154: of  from
9. Discussion is too brief. The authors could slightly extend their Results/Discussion to toxin or to Melioidosis, if possible, and/or to the function/control of host eIF4A after *B. pseudomallei* infection.

Reviewer #3 (Remarks to the Author):

Burkholderia Lethal Factor 1 (BLF1) is an important virulence determinant of Burkholderia pseudomallei and this is the first report to describe the structure of a noncatalytic BLF1 variant complexed with human eukaryotic initiation factor 4A (eIF4A). The structural analysis of the BLF1 C94S:eIF4A complex indicates that BLF1 interacts with the N- and C-terminal RecA-like domains of eIF4A and holds them in a novel twisted conformation. The researchers also assessed the BLF1 active site architecture and compared it to that of other characterized deamidases and proteases with a Cys-His pair at their active site. There will undoubtedly be interest in this work in both the melioidosis and deamidase/protease research communities. There are no concerns with statistical analysis or the ability to repeat the experiments reported in this manuscript. This is impressive work and all that is needed for improvement is to italicize the genus and species names in the references section.

We thank the reviewers for their careful reading of the manuscript and for their suggestions regarding improvements they would like to have made. In the light of their comments we have modified the paper as detailed below:

Reviewer #1 (Remarks to the Author):

- If I have one general criticism about this work it would be that the structural analysis has not led to further experimentation to corroborate conclusions drawn from the structure, not necessarily by performing laboratory experiments but potentially by using high-level computations such as MM/QM simulations to translate the more static crystallographic pictures obtained in this study into trajectories that could provide even more insight into the mechanism of CNF-like deamidases. However, given that this manuscript is a communication and knowing how costly and time-consuming (and frustrating) such calculations can be, I am convinced that this manuscript is a valuable contribution as it stands. – We accept that MM/QM simulations would add to our understanding, however that would be a much longer-term project, and would require the structure of the complex to be determined at a much higher resolution. In particular, the position of all the atoms in the active site together with the surrounding water structure would need to be far more precise than is possible with the current resolution of our structure. We have been endeavouring to find suitable crystallisation conditions but have so far not succeeded in producing the necessary better diffracting crystals. We thank the referee for the comment that the manuscript is a valuable contribution as it stands.
- Line 32: Is the comma after proteases correct? – comma removed
- L118-121: The authors speculate about the protonation state of H106/C94, which they base on geometric considerations. Of course, the very high resolution structure allows for such speculations. There are two questions here: could the authors perform calculations to determine the protonation state of the respective residues (I remember that these are not too difficult to do), and can they (maybe just in a rebuttal letter) provide a bit more insight into how they have refined the high resolution crystal structure of apo BLF1, since refining against such data can provide a challenge in itself but may actually lead to direct observation of protons if one is lucky? – While the determination of the apo BLF1 structure is at very high resolution, unfortunately even higher resolution (better than 1 Å) is required to identify hydrogens. The structure was refined with those hydrogens whose positions are certain (so-called ‘riding’ hydrogens), but they do not include the O, N, or S hydrogen atoms on the histidine, lysine, serine, cysteine, etc. The residual density for non-riding hydrogens was not sufficiently reliable to unambiguously determine their positions in all cases. Therefore, the protonation states of H106 and C94 cannot be determined directly from the X-ray structure.
- L216-217 sound slightly strange, maybe there should be a comma after slightly (this referee is not a native speaker) – reworded
- L238: the comma after because seems wrong – agree, deleted
- L281: using just “TEV” to refer to “TEV protease” seems to be lab jargon – consider revising to be clear about what is meant – We had defined the use of the ‘TEV’ abbreviation on line 404. The referee may have overlooked this.
- L376-377: the authors speculate that BLF1 mimics binding of RNA to eIF4A – are there structures to support this? – The involvement of arginines being involved in binding RNA was discussed earlier in the manuscript, but we have altered the discussion to make it clearer, see lines 300-332, and 554-558.
- In table 1, I am missing information about CC1/2 – CC1/2 has been added to table 1
- Stereo figures: I suggest changing from wall-eyed to cross-eyed layout. This may be a subjective suggestion, but I have difficulties to perceive figures in this manuscript in stereo (I

double-checked with both options in PyMOL to write this report, cross-eyed seems better (for me at least)). - As suggested by the referee we have switched to cross-eyed stereo and adjusted the labels to improve the stereo depth. This applies to figures 1A, 4A, 4C, 4E, and S3.

- I also have the impression that the labels within the stereo figures are not adjusted to the depth of the figure, this could be improved. - See above
- L580: refinement of the apo structure – I wonder if REFMAC will really give the best results here, have the authors also tried other programs (SHELXL etc)? – The method of refinement is somewhat of a personal choice, particularly where good results are achieved. We note that the eIF4A:BLF1 form A is within the top 3 % of all structures at this resolution deposited in the RCSB, consistent with the refinement being well done.

Reviewer #2 (Remarks to the Author):

- Please provide broader meaning/speculation of this research at the end of the abstract. – We have extensively reworded the abstract
- As a non-native but scientific-English-fluent scientist, this reviewer asks the authors to revise complex sentences and relatively long sections to make them more clear and intuitive in meaning. Some of them are as follows:
 - line 62-67 (too long and complex for readers to easily follow and understand) – Sentence has been re-worded to make it easier to read
 - line 87-89 – sentence re-structured
 - line 161-227 (could be divided into two sections), - The section has been divided and re-ordered to improve clarity. See also changes made to this section regarding RNA in response to reviewer 1.
 - line 233-237 (too long and complex for readers to easily follow and understand), - sentence divided into two and re-worded
 - line 271-368 (could be divided into 2~3 sections) – We thank the referee for this helpful comment and have divided this section into three.
 - line 307-310. – Re-worded and moved in the text to improve clarity
- In the beginning of Introduction, brief/short description of Melioidosis symptoms and pathogenic severeness would be informative for readers to be able to roughly understand the disease without further consideration. – An extra section to the introduction has been added to address this comment (see lines 109 – 115)
- Line 38: man  human – changed the sentence to use the word humans
- Line 66: one another  each other(???) – sentence modified as suggested
- Line 97: between BLF1 and CNF1, their structures commonly adopts a beta-sandwich fold and could be well superimposed with r.m.s.d. of ?? angstrom among all the Calpha atoms (This part could be revised something like this). – sentence has been modified and the RMSD value between the two structures has been added.
- Line 135-137: Figure S2A could be placed in the manuscript since it shows the major difference between two forms. – We agree with the referee that this figure does show the detailed difference between our two crystal forms, but since the differences are only subtle and since figure 1 is very similar to S2A, to avoid repetition and to save space in the journal, figure S2A was relegated to the supplementary information and we remain of the view this is appropriate.
- Line 154: of  from - not sure which one they are referring to - changed
- Discussion is too brief. The authors could slightly extend their Results/Discussion to toxin or to Melioidosis, if possible, and/or to the function/control of host eIF4A after B. pseudomallei infection. – We have increased the discussion by ~ 50 % following the suggestions by the

referee of the type of content that could be extended.

Reviewer #3 (Remarks to the Author):

- This is impressive work and all that is needed for improvement is to italicize the genus and species names in the references section. – **References have been italicised where appropriate**